# Social Support and Mental Health in the Postpartum Period in Times of SARS-CoV-2 Pandemic: Spanish Multicentre Cohort Study

**DOI:** 10.3390/ijerph192315445

**Published:** 2022-11-22

**Authors:** Maia Brik, Miguel Sandonis, Elena Rocio Horrillo Murillo, Rogelio Monfort Ortiz, Alexandra Arteaga Fernandez, Maria de Arriba, Sara Fernández, Nuria Iglesias Román, Gemma Parramon-Puig, Anna Suy, María Emilia Dip, Alfredo Perales Marin, Nerea Maiz, Josep Antoni Ramos-Quiroga, Elena Carreras

**Affiliations:** 1Maternal-Fetal Medicine Unit, Department of Obstetrics, Hospital Universitari Vall d’Hebron, Universitat Autònoma de Barcelona, 08035 Barcelona, Spain; 2Department of Mental Health, Hospital Universitari Vall d’Hebron, 08035 Barcelona, Spain; 3Department of Psychiatry and Forensic Medicine, Universitat Autònoma de Barcelona, 08035 Barcelona, Spain; 4Obstetrics Department, Hospital Universitario 12 de Octubre, 28041 Madrid, Spain; 5Obstetrics Department, Hospital Universitario La Fe, 46026 Valencia, Spain; 6Group of Psychiatry, Mental Health and Addictions, Vall d’Hebron Research Institute (VHIR), 08035 Barcelona, Spain; 7Biomedical Network Research Centre on Mental Health (CIBERSAM), 08035 Barcelona, Spain

**Keywords:** anxiety, depression, pregnancy, pandemic, social support, COVID-19, SARS-CoV-2

## Abstract

Background: To explore the depression and anxiety symptoms in the postpartum period during the SARS-CoV-2 pandemic and to identify potential risk factors. Methods: A multicentre observational cohort study including 536 women was performed at three hospitals in Spain. The Edinburgh Postnatal Depression Scale (EPDS), the State-Trait Anxiety Inventory (STAI) Scale, the Medical Outcomes Study Social Support Survey (MOS-SSS), and the Postpartum Bonding Questionnaire (PBQ) were assessed after birth. Depression (EPDS) and anxiety (STAI) symptoms were measured, and the cut-off scores were set at 10 and 13 for EPDS, and at 40 for STAI. Results: Regarding EPDS, 32.3% (95% CI, 28% to 36.5%) of women had a score ≥ 10, and 17.3% (95% CI, 13.9% to 20.7%) had a score ≥ 13. Women with an STAI score ≥ 40 accounted for 46.8% (95% CI, 42.3% to 51.2%). A lower level of social support (MOS-SSS), a fetal malformation diagnosis and a history of depression (*p* = 0.000, *p* = 0.019 and *p* = 0.043) were independent risk factors for postpartum depression. A lower level of social support and a history of mental health disorders (*p* = 0.000, *p* = 0.003) were independent risk factors for postpartum anxiety. Conclusion: During the SARS-CoV-2 pandemic, an increase in symptoms of anxiety and depression were observed during the postpartum period.

## 1. Introduction

The SARS-CoV-2 pandemic resulted in a global health crisis. Pregnancy and postpartum are critical periods for mental health. Stressors during pregnancy, such as traumatic psychological events, low socio-economic status, and the presence of depression and anxiety, are associated with poorer obstetric and infant outcomes [1].

The prevalence of anxiety disorder in the general population is 13.6% [2], which increases to 15.2% during pregnancy [3]. The prevalence of depression in the general population is 2.8%, increasing to 7.4–12.8% during pregnancy [4]. During the strict lockdown period of March to May 2020 in Spain, the rate of anxiety symptoms in pregnant women increased by 59% and the rate of depression symptoms increased by 39% [5]. Therefore, an increase in the rates of both depression and anxiety during the postpartum period is likely.

Postpartum stress is negatively associated with poor developmental trajectories and linear growth deficits, causing stunted growth; poor language and cognitive development; poor gross and fine motor movement, and infant sleep [6]. An inverse relationship exists with breast feeding and postpartum depression [6]. Postpartum depression impairs maternal functioning because it worsens maternal nutrition, interferes with children breastfeeding and surveillance, and decreases maternal–infant bonding [6].

Identifying the risk factors for postpartum depression and anxiety during pregnancy will help to improve the antenatal care of women who present a higher risk for mental health disorders during the postpartum period, and help to prevent such disorders. 

The main aim of the present study was to analyze potential depression and anxiety symptoms, as well as the level of mother-to-infant bonding, in the postpartum period of women who were pregnant during the SARS-CoV-2 pandemic. A secondary aim was to identify risk factors for these symptoms.

## 2. Material and Methods

We performed a multicentre observational cohort study, including women who delivered after the SARS-CoV-2 pandemic outbreak, in three tertiary hospitals in Spain (Hospital Universitari Vall d’Hebrón in Barcelona, Hospital 12 de Octubre in Madrid, and Hospital La Fe in Valencia). Recruitment lasted from April 2020 to December 2020. 

Inclusion criteria included women delivering a live newborn after the SARS-CoV-2 pandemic outbreak, able to read and/or communicate in Spanish and having a personal e-mail account. Women were invited to participate, either in person or by telephone, during the first month after birth.

The primary outcomes included the presence of depression and anxiety symptoms, and mother-to-infant bonding, as measured by several questionnaires administered by e-mail. The Edinburgh Postnatal Depression Scale (EPDS), the State-Trait Anxiety Inventory (STAI) and the Postpartum Bonding Questionnaire (PBQ), measure levels of depression, anxiety, and mother-to-infant bonding, respectively. A questionnaire to assess social support, the Medical Outcomes Study Social Support Survey (MOS-SSS) was administered.

During the data-collection period, many women completed one or more of the questionnaires. All fully completed questionnaires were included in the study.

The EPDS is a self-reported scale designed to detect postpartum depression. It consists of 10 self-administered items, each rated on a 4-point scale ranging from 0 to 3, with higher scores indicating a greater severity [7]. The best cut-off score of the Spanish version of the EPDS was 10/11, with a sensitivity of 79%, a specificity of 95.5%. For the EPDS, a cut-off score of 13 has a sensitivity of 62% and a specificity of 98.1% [8]. In the EPDS questionnaire, item number 10 evaluates suicidal ideation, and was examined independently of the EPDS score.

The State-Trait Anxiety Inventory (STAI) questionnaire is the main rating scale for the state (STAIs) and trait (STAIt) anxiety [9]. It consists of 40 self-administered items rated on a 4-point scale, ranging from 0 to 3. The STAI scale has also been validated for use in pregnant women [10]. The scores for each subtest range from 20 to 80, with higher scores indicating a greater anxiety. A cut-off score of 39–40 in the STAI scale has been suggested as being indicative of clinically significant symptoms of anxiety.

The PBQ questionnaire is a 25-item self-administered rating scale to evaluate the level of mother-to-infant bonding. It is commonly used to identify problems with general bonding, rejection and pathological anger, and anxiety about infant and incipient abuse. An overall PBQ score of 26 or higher has been suggested as being indicative of mother-to-infant bonding disorder [11].

The MOS-SSS questionnaire is a 20-item self-administered rating scale developed to measure the level of social support (positive social interaction, as well as tangible, affectionate, and emotional/informational support) [12].

Study data were collected from the electronic system of medical records, and managed using REDCap™ electronic data capture tools hosted at VHIR-HOSPITAL UNIVERSITARI VALL DE HEBRON [13]. EPDS, STAI and MOS-SSS questionnaires were also sent by this system at 4 weeks postpartum. Women completed the questionnaires between 4 and 8 weeks postpartum. The PBQ questionnaire was sent at 3 months postpartum.

In order to assess potential risk factors for depression and anxiety, several variables were recorded: maternal characteristics (maternal age, pre-pregnancy BMI, pre-pregnancy maternal weight, low-risk pregnancy, history of mental health disorders, history of depression, parity, assisted reproductive technique, ethnicity, smoking during pregnancy, SARS-CoV2 infection during pregnancy, psychotropic medication during pregnancy), complications during pregnancy (gestational diabetes, pregnancy-induced hypertension, fetal growth restriction, fetal malformation, cesarean section, preterm birth <37 weeks), neonatal outcomes (breastfeeding at discharge, admission to ICU, cord gases, birth weight). Low-risk pregnancy was defined according to the “Protocol de seguiment de l’embaràs. Agencia de Salut Pública de Catalunya”. History of mental health disorder was defined occur before pregnancy or during the current pregnancy and stated by a mental health clinician. Assisted reproductive techniques included in vitro fertilization or artificial insemination pregnancy. Fetal growth restriction was defined if estimated fetal weight was below the 10th centile during pregnancy, measured by ultrasound and using Hadlock chart [14].

Sampling methods: we offered women the option to participate either by telephone or face-to-face before hospital discharge after delivery. For the telephonic recruitment, the birth register was searched to select the women who would be contacted, and they were called by phone regarding participation in the study. For those participants who agreed, an electronic consent form and the study questionnaires were sent by the REDCap™ system. For the face-to-face recruitment, a round was carried out in the postpartum ward to recruit women for the study and a written consent form was obtained on-site, accepting the study questionnaires sent by the REDCap™ system.

Assuming an alpha error of 0.05, and a beta error of 0.2 in a bilateral contrast, it was estimated that the number of subjects necessary to detect a difference equal or superior to 0.05 units was 351. A proportion of 0.1 in the reference group of postpartum depression [15] and a 10% rate of loss to follow-up were assumed.

### Statistical Methods

Continuous variables were expressed as the median and interquartile range (IQR) or the mean and standard deviation (SD). Categorical variables were expressed as frequency and percentage. A univariate linear regression analysis was performed to identify risk factors for depression and anxiety symptoms. A correlation analysis between MOS-SSS scores and EPDS, STAI and PBQ scores was performed.

The IBM SPSS Statistics Software, version 23 (IBM Corp., Armonk, NY, USA), and the R software were used for the statistical analyses. All reported probability values were two-tailed, and significance was set at *p* = 0.05.

## 3. Results

A total of 1193 women were offered the option to participate in the study, 697 women were enrolled in the study and 547 completed at least one of the questionnaires (EPDS, STAI, MOS-SSS, and PBQ). Therefore, the response rate was 78.4% (547 out of 697). From 547 women, 467 (85.3%) responded to the EPDS questionnaire, 485 (88.6%) to the STAI questionnaire, 532 (97.2%) to the MOS-SSS questionnaire, and 337 (61.6%) to the PBQ questionnaire. Due to incomplete data, 428 women were included for the risk factor analysis (Figure 1).

### 3.1. Descriptive Data

Table 1 describes maternal demographics, along with pregnancy, delivery and neonatal. Of the 428 women included in the risk factor analysis, 425 (99.3%) completed the EPDS questionnaire, 425 (99.3%) completed the STAI questionnaire, 425 (99.3%) completed the MOS-SSS questionnaire, and 292 (68.2%) completed the PBQ questionnaire. 

Outcome data (*n* = 547)

Table 2 describes the outcome descriptive data. Regarding the EPDS questionnaire, 151 out of 467 women (32.3%, 95% CI, 28% to 36.5%) had a score ≥ 10, and 81 out of 467 (17.3%, 95% CI, 13.9% to 20.7%) had a score ≥13. Regarding suicidal ideation (EPDS item 10), 25 out of 467 (5.0%, 95% CI, 3% to 7%) responded positively to this question. 

Regarding the STAI scale, 238 out of 485 (46.8%, 95% CI, 42.3% to 51.2%) had a score ≥ 40; and 231 out of 485 (47.3%, 95% CI, 43.1% to 52%) had a score ≥ 40.

Regarding the PBQ scale, 40 out of 337 women (11.8%, 95% CI, 8.4 to 15.3%) had a score ≥ 26.

### 3.2. Regression Analysis for Depression Symptoms

A univariate linear regression analysis was performed to identify potential risk factors for postpartum depression symptoms using the EPDS questionnaire. Having a history of depression, caucasian ethnicity, a lower level of social support (MOS-SSS) and the presence of fetal malformation (*p* = 0.043, *p* = 0.038, *p* = 0.000, *p* = 0.019) were identified as independent predictive risk factors for postpartum depression (Table 3). 

### 3.3. Regression Analysis for Anxiety Symptoms

A univariate linear regression analysis was performed to identify demographic, pregnancy, delivery and neonatal variables as potential risk factors for anxiety symptoms in the STAIs questionnaire. Lower social support (MOS-SSS) and having a mental health disorder during pregnancy were identified as predictive factors for higher anxiety symptoms (*p* = 0.000, *p* = 0.003), (Table 3).

Level of social support was assessed as a variable associated with mental health disorders during the postpartum period. An inverse correlation between MOS-SSS score and EPDS, STAIs, STAIt and PBQ scores was observed (r = −0.362, r = −0.402, r = −0.411, r = −0.217, respectively, *p* = 0.000).

There was an inverse correlation between different MOS-SSS items (emotional/informational support, tangible support, positive social interaction and affectionate support) and EPDS, STAIs, STAIt and PBQ scores (Table 4).

### 3.4. Mother-to-Infant Bonding Disorder and Maternal Mental Health

When examining PBQ scores against EPDS and STAIs scores, an increased risk of mother-to-infant bonding disorder (PBQ score above 26) was observed for cases with an EPDS score ≥ 10 or ≥13, and a STAIs score ≥40. PBQ scores against EPDS and STAIs scores are shown in Table 5.

## 4. Discussion

The present study suggests an increased risk of anxiety and depression symptoms [15,16] during the postpartum period, and an increased risk of mother-to-infant bonding disorder during pandemic. Furthermore, the presence of mental health disorders was predictive of postpartum anxiety symptoms, and a history of depression was predictive of postpartum depression. Finally, a low level of social support leads to increased levels of both anxiety and depression symptoms during the postpartum period, and to an increased risk of mother-to-infant bonding disorder. 

The main strengths of the study are a quantitative assessment of the social support, and an in-depth study on the interaction between the impact of COVID-19 on both depression and anxiety during the postpartum period and the level of social support. 

In addition, the on-line completion of the questionnaires is a strength, since women had time to complete them properly. However, it could also represent a selection bias due to some women not having Internet connection or devices to access the Internet, generally those with lower social support. Another limitation is the language selection bias because only women fluent in Spanish were offered the chance to participate. However, this enhances the reliability of the results, since the women fully understood the purpose and the implications of the study. 

The present study suggests a significant increase in the risk of postpartum depression as compared to previous data in pregnant women within the Spanish population, with 10.4% of women having an EPDS score ≥10 in the postpartum period before the pandemic [15]. In the literature, the prevalence of depression symptoms in the postpartum period during the SARS-CoV-2 pandemic is between 14% and 59% [17,18]. These results are in agreement with the present study, where EPDS scores ≥10 and ≥13 indicated prevalence rates of depression symptoms of 32% and 17%, respectively.

A previous meta-analysis study found a non-significant increase in depression symptomatology during pregnancy and the postpartum period during the pandemic compared to before the pandemic [16]. On the other hand, studies comparing EPDS scores during vs. before the pandemic do not show consistent results [19,20]. The differences across these studies can be explained by several factors: different cut-off points for EPDS scores (10 or 13), including both pregnant and postpartum women, the different prevalence of postpartum depression before the pandemic, and the different moments of recruitment.

As regards to depression symptoms during the postpartum period, the correlation before and during the pandemic is reported to be around one third in the literature [20], and these data are consistent with our results. EPDS scores ≥ 10 postpartum was 32% in our study, whereas the prevalence of postpartum depression in the pre-pandemic period in the Spanish population was 10.4% [15].

Regarding anxiety in the postpartum period, and according to the STAI score, an increase in the STAI score in pregnant and postpartum women during the pandemic has also been reported in the literature [16].

Moreover, the present study found that pregnant women with a history of mental health disorders had better scores for postpartum anxiety symptoms during the pandemic, and this had already been reported in the literature [5]. These results may be explained by the fact that women with a history of mental health disorders could have developed increased resilience, and, thus, a history of mental health disorders may act as a protective factor [21]. However, this result is not consistent with previous studies [22]. In the Liu C et al. study, maternal mental health history was self-reported when completing the questionnaires during the pandemic, whereas in our study it was determined by clinical interview with a mental health consultant before the pandemic. On the other hand, all pregnant women with mental health disorders were followed at the perinatal mental health clinic, and this highlights the fact that proper mental health support in these women acts as a preventive factor for anxiety and depression symptoms in the postpartum period.

Furthermore, a low level of social support has been reported in the literature as a risk factor for both anxiety and depression in the postpartum period. Living in areas with a low socio-economic status during the pandemic has been correlated with higher EPDS scores during the postpartum period and, therefore, a higher risk of postpartum depression [23]. However, in our study, the level of social support was individually measured by the MOS-SSS questionnaire, which examines how a woman, as an individual, has been socially supported. Thus, according to the literature, social support has the potential to buffer against adverse mental health outcomes, particularly in those more negatively affected by the SARS-CoV-2 pandemic [24]. This suggests that strategies to provide social support may help to prevent and treat postpartum depression and anxiety. 

One of the consequences of postpartum depression and anxiety is the mother-to-infant bonding disorder, and this has been reported to be present in 7.1 to 8.6% of women in the general population [25]. In the present study, the rate of mother-to-infant bonding disorder (PBQ score above 26) was 11.8%, which seems to be in agreement with the increased rate of depression symptoms during pregnancy and the postpartum period during the pandemic [5].

The suicidal ideation rate in our population was 5%, which is consistent with the overall suicidal ideation rate of around 5–14% during pregnancy and the postpartum period in a study conducted before the pandemic [26].

The findings of the study can be applied to populations with a similar prevalence of depression and anxiety during the postpartum period and similar healthcare systems to that of Spain. 

This study highlights the relevance of an active screening for mental health disorders risk factors during pregnancy and a follow-up treatment based on cooperation between the obstetrics and mental health teams, as they are the major methods of postpartum depression prevention.

## 5. Conclusions

To conclude, the SARS-CoV-2 pandemic might have an impact on the risk of depression and anxiety symptoms during the postpartum period. In addition, women with low levels of social support are at a higher risk of both depression and anxiety during this period. 

## Figures and Tables

**Figure 1 ijerph-19-15445-f001:**
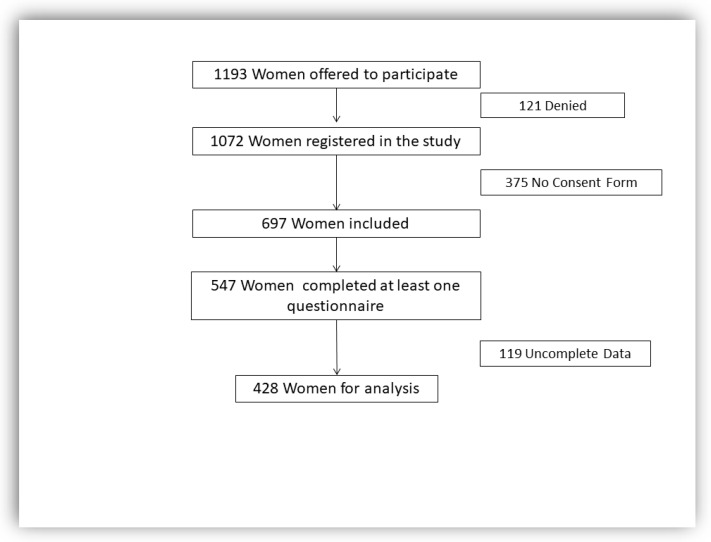
Flow Diagram.

**Table 1 ijerph-19-15445-t001:** Maternal demographics, and pregnancy, delivery and neonatal outcomes (*n* = 428).

Maternal Demographics	
Maternal age (y), mean ± SD	33.6 ± 5.8
Pre-pregnancy BMI, mean ± SD	25.1 ± 4.9
Pre-pregnancy maternal weight (kg), mean ± SD	66.9 ± 13.9 (65.6–68.5)
Low risk pregnancy *	306 (71.4% (76–67))
History of mental health disorders *	20 (4.6% (3–7%))
History of depression *	6 (1.4% (0–3))
Nulliparous *	227 (53% (49–59))
Assisted reproductive technique *	29 (6.7% (4–9%))
Ethnicity	
Caucasian *	328 (76.6% (72.7–80.9))
Latin-American *	76 (17.8% 14.3–21.7))
Smoking status *	28 (6.5% (4–9%))
SARS-CoV-2 infection during pregnancy *	27 (6.3% (4–9))
Psychotropic medication during pregnancy *	9 (2.1% (1–3%))
**Pregnancy, delivery and neonatal outcomes**	
Gestational diabetes *	32 (8.3% (4–10%))
Pregnancy-induced hypertension *	14 (3.6% (2–7%))
Fetal growth restriction *	26 (6.7% (4–10%))
Fetal distress during labour *	32 (8.3% (6–11))
Fetal malformation *	10 (2.6% (1–5%))
Cesarean section *	102 (26.4% (22.2–30.5))
Preterm birth <37 weeks *	20 (5.2% (3.7–8.4))
Breastfeeding at discharge *	322 (83.4% (89–94))
Admission to the neonatal ICU *	22 (5.7% (4–9%))
Arterial cord pH, mean ± SD	7.22 ± 0.08
Venous cord pH, mean ± SD	7.30 ± 0.06
Birth weight (grams), mean ± SD	3265 ± 504

* Number of women (% (95% CI)).

**Table 2 ijerph-19-15445-t002:** Descriptors of depression (EPDS) and anxiety (STAI) symptoms, social support (MOS-SSS), and infant-to-mother bonding (PBQ).

EPDS Score, Mean ± SD	7.4 ± 5
EPDS ≥ 10 *	151/467 (32.3% (28–36.5))
EPDS ≥ 13 *	81/467 (17.3% (13.9–20.7))
STAIs questionnaire, mean ± SD	39.9 ± 11.7
STAIs ≥ 40 *	238/485 (49.0% (42.3–51.2))
STAIt questionnaire, mean ± SD	40.2 ± 10.9
STAIt ≥ 40 *	231/485(47.6% (43.1–52))
Total MOS-SSS score, mean ± SD	78.5 ± 17
Emotional-informational support	32.8 ± 7.5
Tangible support	16.1 ± 4.2
Affectionate support	16.5 ± 3.8
Positive social interaction	13.0 ± 2.5
Total PBQ score, mean ± SD	16.3 ± 10
PBQ Factor 1. General Bonding Factor	10.7 ± 4.8
PBQ Factor 2. Rejection and pathological anger	1.6 ± 2.7
PBQ Factor 3. Anxiety about the infant	3.4 ± 2.6
PBQ Factor 4. Incipient abuse	0.1 ± 0.6
PBQ ≥ 26 *	40/337 (11.8% (8.4–15.3))

* Expressed as absolute values/total (percentage (95% CI)).

**Table 3 ijerph-19-15445-t003:** Univariate linear regression logistic analysis for depression (EPDS) and anxiety (STAIs) symptoms, and demographic, pregnancy, delivery and neonatal variables (*n* = 428).

	Depression Symptoms (EPDS)			Anxiety Symptoms (STAIs)		
	B	95% CI	*p*	B	95% CI	*p*
Maternal Demographics						
Maternal Age	0.045	−0.107 to 0.282	0.379	0.045	−0.107 to 0.282	0.379
BMI	−0.019	−0.128 to 0.091	0.738	−0.004	−0.266 to 0.249	0.948
Weight	−0.031	−0.048 to 0.026	0.569	−0.004	−0.090 to 0.083	0.934
High-risk pregnancy	−0.036	−1.464 to 0.683	0.475	−0.034	−3.359 to 1.642	0.500
History of mental health disorders	−0.083	−4.146 to 0.376	0.102	−0.147	−12.925 to −2.622	0.003
History of depression	0.482	0.146 to 7.670	0.043	0.109	−9.077 to 14.000	0.658
Parity	−0.088	−1.868 to 0.114	0.083	0.013	−1.991 to 2.589	0.798
ART	0.025	−1.423 to 2.375	0.622	0.027	−3.268 to 5.712	0.593
Caucasian	−0.247	−5.650 to −0.167	0.038	−0.150	−10.571 to 2.334	0.210
Latin-American	0.379	−4.236 to 1.617	0.379	−0.094	−9.753 to 4.000	0.411
Smoking status	−0.004	−2.001 to 1.846	0.937	0.059	−1.862 to 7.055	0.253
SARS-CoV-2-infection during pregnancy	0.045	−1.108 to 2.907	0.379	0.024	−3.483 to 5.738	0.631
MOS-SSS questionnaire	−0.374	−1.139 to −0.084	0.000	−0.379	−0.329 to −0.202	0.000
Pregnancy, Delivery and Neonatal outcomes						
Gestational diabetes	0.002	−1.795 to 1.874	0.967	0.200	−3.390 to 5.129	0.689
Pregnancy-induced hypertension	0.076	−0.564 to 4.144	0.136	0.082	−0.915 to 10.037	0.102
Fetal growth restriction	0.024	−1.470 to 2.386	0.640	−0.018	−5.304 to 3.639	0.715
Preterm birth	−0.023	−2.245 to 1.541	0.654	−0.037	−6.391 to 2.876	0.456
Fetal malformation	−0.119	−5.471 to −0.501	0.019	−0.092	−11.272 to 0.338	0.065
Maternal weight gain	0.054	−0.048 to 0.129	0.367	2.337	−0.655 to 5.321	0.125
Cesarean due to fetal distress	0.029	−1.267 to 2.326	0.563	0.017	−3.478 to 4.908	0.738
Arterial cord pH	−0.049	−0.003 to 0.001	0.378	−0.059	−0.007 to −0.002	0.275
Venous cord pH	0.007	−7.791 to 8.858	0.900	0.052	−9.517 to 28.484	0.327
Birth weight	0.035	−0.001 to 0.001	0.699	−0.034	−0.001 to 0.003	0.500
Birth weight centile	−0.027	−0.023 to 0.014	0.610	−0.003	−0.044 to 0.042	0.942
Breastfeeding	−0.028	−2.381 to 1.343	0.584	−0.013	−4.691 to 3.600	0.796
Admission to the NICU	−0.007	−1.993 to 1.729	0.890	0.000	−4.349 to 4.334	0.997

NICU: Neonatal Intensive Care Unit. ART: Artificial Reproductive Technique.

**Table 4 ijerph-19-15445-t004:** Correlation between the MO-SSS score and EPDS, STAIs, STAIt and PBQ scores (r = Pearson).

	MOSS Questionnaire	Emotional-Informational Support	Tangible Support	Positive Social Interaction	Affectionate Support	*p*
STAIs	−0.402	−0.366	−0.328	−0.426	−0.371	0.000
STAIt	−0.411	−0.386	−0.319	−0.423	−0.389	0.000
PBQ	−0.217	−0.218	−0.150	−0.208	−0.201	0.000
EPDS	−0.362	−0.332	−0.293	−0.370	−0.351	0.000

**Table 5 ijerph-19-15445-t005:** Mother-to-infant bonding (PBQ) in relation to depression and anxiety symptoms (EPDS and STAIs) at different cut-off scores.

	EPDS			EPDS			STAIs		
	≥10	<10	*p*	≥13	<13	*p*	≥40	<40	*p*
PBQ ≥ 26	24/86 (27.9%)	8/213 (3.8%)	0.000	15/50 (30%)	17/249 (6.8%)	0.000	27/138 (19.6%)	6/168 (3.6%)	0.000

## Data Availability

Data available on request due to restrictions (ethical consent).

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
