# Peer review of "Social Support and Mental Health in the Postpartum Period in Times of SARS-CoV-2 Pandemic: Spanish Multicentre Cohort Study"

_ijerph, 2022, doi:10.3390/ijerph192315445_

Round 1

Reviewer 1 Report

This is a well written manuscript that reports on the prevalence of anxiety and depression amongst women living in Spain in the postpartum period after the Covid-19 outbreak and highlights some mechanisms that might increase risk of these conditions. It contributes to the evidence base about the importance of social support for women's wellbeing in the postpartum period, in particular.

I have one main query:

We know that people /women in lower socio-economic groups have been adversely affected by the pandemic which has contributed to already exiting health inequalities. Did you measure for socio-economic status in your study (i.e. by income or education or area)? Did you consider whether this would be relevant to women's outcomes? Please could you address this, and note it as a limitation if it was not explored.  

There are some minor typos, I have noted a couple: 

p2 - line 63 - it should read live newborn 

p3 - line 108 - it should read electronic consent form

Author Response

Dear Reviewer,

Thank you for your comments. Please find attached the comments to your revision of the Manuscript: " Social support and Mental Health in the Postpartum Period in times of SARS-CoV-2 pandemic: Spanish Multicentre Cohort Study"

Sincerely,

Dr Maia Brik

Reviewer 2 Report

This study explored the depression and anxiety symptoms in the postpartum period 16 during the SARS-CoV-2 pandemic and identified potential risk factors.

Introduction part

The reviewer suggests some statements need to add citations or references. For example, “postpartum stress is negatively associated with poor developmental trajectories and linear growth deficits, causing stunting in growth; poor language and cognitive development; poor gross and fine motor movement, and infant sleep” need to add relevant references.

Material and Methods part

In this study, the author collected plenty of important variables and demonstrated many statistics in the results. However, these variables didn’t clearly address the definition and the response to these questions. The reviewer suggests the authors need to add a detailed statement or account of these variables such as what is low-risk pregnancy, history of mental health disorders, assisted reproductive technique, and smoking status (please tell us the study just included current smokers in your study and excluded ex-smoker), fetal growth restriction, fetal distress during labor…and so on. The detailed definition can help readers understand your study design and can distinguish the difference between this study and others.

The reviewer didn’t find the contents of Appendix 1 and the manuscript didn’t mention IRB’s information.

This study design was a cohort study and adopted face-to-face, telephone, and online questionnaires three methods to collect data. It seems so complex to understand how to access the participants. The reviewer suggests the authors need to state this part more clearly.

Results part

1193 women were offered to participate in the study, but 697 women were enrolled 124 in the study, and 547 completed at least one of the questionnaires (EPDS, STAI, MOS-SSS, 125, and PBQ). Therefore, the response rate was 78.6%. Please explain how to calculate 78.6%.

Page 5 (line 139)

The authors stated outcome data adopted 527 to analysis. However, the authors mentioned 547 completed at least one of the questionnaires in the results part. The authors need to confirm it.

Page 6

Caucasian was a significant variable in univariate linear regression logistic analysis for depression. The authors didn’t mention it. The reviewer suggests explaining it.

Discussion part

The authors stated that “The present study suggests an increased risk for anxiety and depression symptoms (14,15), during the postpartum period, and an increased risk for mother-to-infant bonding disorder during the pandemic”. Why do the findings of the present study need to cite other studies?

(#14: Garcia-Esteve L, Ascaso C, Ojuel J, Navarro P. Validation of the Edinburgh Postnatal Depression Scale (EPDS) in Spanish mothers. J Affect Disord. 2003 Jun;75(1):71–6. #15: Hessami K, Romanelli C, Chiurazzi M, Cozzolino M. COVID-19 pandemic and maternal mental health: a systematic review and meta-analysis. J Matern Fetal Neonatal Med. 2022 Oct;35(20):4014–21.?)

The authors stated that the “Main strengths of the study are the inclusion of the level of social support as a potential variable since it highlights social support is a contributing factor to postpartum mental health”. However, previous studies revealed that social support was an important predictor of postpartum mental health. Please explain why the inclusion of the level of social support in this study is an advantage.

Author Response

(The authors gave the same response as above.)

Round 2

Reviewer 2 Report

No comments.